# Evaluation of Pulmonary Hypertension in Dogs with Heartworm Disease Using the Computed Tomographic Pulmonary Trunk to Aorta Diameter Ratio

**DOI:** 10.3390/ani12182441

**Published:** 2022-09-16

**Authors:** Jorge Isidoro Matos, Yaiza Falcón-Cordón, Sara Nieves García-Rodríguez, Noelia Costa-Rodríguez, José Alberto Montoya-Alonso, Elena Carretón

**Affiliations:** Internal Medicine, Veterinary Medicine and Therapeutic Research Group, Faculty of Veterinary Medicine, Research Institute of Biomedical and Health Sciences (IUIBS), Universidad de Las Palmas de Gran Canaria, 35016 Las Palmas de Gran Canaria, Spain

**Keywords:** heartworm disease, *Dirofilaria immitis*, pulmonary hypertension, computed tomography, pulmonary trunk to aorta ratio, dogs, animal diseases

## Abstract

**Simple Summary:**

Pulmonary hypertension (PH) is a consequence of proliferative pulmonary endoarteritis caused by the parasite *Dirofilaria immitis* (heartworm). To date, the diagnosis is carried out through echocardiographic determinations, with the right pulmonary artery distensibility (RPAD) index being the gold standard for diagnosis of PH caused by heartworm. Given the high importance and frequency of PH in dogs, the study of the pulmonary trunk to aorta ratio (PT:Ao) determined by computed tomography (CT) was proposed to assess its usefulness in the diagnosis of PH, compared to the gold standard method. The results show that the PT:Ao ratio was reliable for detecting and staging the severity of moderate to severe PH in dogs with heartworm.

**Abstract:**

*Dirofilaria immitis* causes proliferative pulmonary endoarteritis that leads to the appearance of chronic precapillary pulmonary hypertension (PH) in dogs. Pulmonary trunk to aorta ratio (PT:Ao ratio) obtained by computed tomography (CT) was studied and the quantitative measure of the diameters of the pulmonary trunk (PT), the descending thoracic aorta (DAo) and ascending thoracic aorta (AAo) were evaluated for the determination of the presence of moderate to severe PH in 59 dogs. The diagnosis of PH was echocardiographically determined, based on the determination of the right pulmonary artery distensibility (RPAD) index (<29.5%), and compared with other parameters for estimating PH. The results showed a very high concordance: 0.976 (*p*-value 0.000) between the two CT methods (PT:DAo and PT:AAo) with an excellent intraclass correlation coefficient > 0.95. Moreover, cut-off values of ≥1.111 for PT:DAo, and ≥1.057 for PT:AAo were determined for dogs with an RPAD index < 29.5%, which suggests a cut-off value between healthy dogs and the presence of PH. As has been previously published, The PT:Ao ratios did not determine the presence of mild PH, so the measurements cannot be considered useful for the early diagnosis of PH in dogs with heartworm.

## 1. Introduction

Canine heartworm disease (HWD) is a severe worldwide vector-borne disease caused by *Dirofilaria immitis*, which mainly affects the pulmonary arteries and the lung parenchyma. The presence of adult parasites in direct contact with the vasculature produces proliferative endoarteritis that chronically leads to the appearance of precapillary pulmonary hypertension (PH), a frequent and serious phenomenon in HWD [1,2]. In addition, the sudden death of the parasites produces pulmonary thromboembolisms that worsen the vascular damage and presence of PH [3,4]. It has been hypothesized that the determination of PH may be highly useful to determine the severity of pulmonary endarteritis [3,5]. Moreover, heartworm-infected dogs with PH show important clinical signs, including cough, exercise intolerance, respiratory distress, and syncope, while right-sided congestive heart failure may occur in advanced stages. Consequently, the objective determination of PH is important [4].

Catheterization of the right side of the heart is considered the gold standard for measuring pulmonary artery pressure and to definitively diagnose PH [6]. However, it involves a high risk and expensive costs, not affordable in veterinary medicine. Thus, diagnostic imaging techniques, specifically echocardiography, provide the most accurate, non-invasive, available, and cost-effective estimate of pulmonary pressure in dogs [7]. Although echocardiography is the method of choice to estimate the presence or absence of PH, in veterinary practice, helical thoracic computed tomography (CT) is becoming more widely available and used more frequently for the evaluation of cardiorespiratory diseases in small animals [8]. Moreover, recent studies have shown the usefulness of CT in the diagnosis of PH in the canine species [9,10,11]; although, to date, there are no published specific indicators of PH in dogs with heartworm.

The use of angiography through CT and the quantitative measure of the diameters of the pulmonary trunk (PT) and the aorta (Ao), translated into the PT:Ao ratio, constitutes a clinical parameter routinely used to aid in the diagnosis of PH in humans, which has also demonstrated an objective value in veterinary medicine in comparison with echocardiography [9,10,11,12,13,14]. The recent use of the PT:Ao ratio has shown that dogs with moderate and severe PH had significantly higher PT:Ao ratios than healthy dogs and dogs with mild PH in a study carried out with dogs with idiopathic pulmonary fibrosis [11]. Furthermore, there are two different measurement methods of the PT:Ao ratio, which are based on the size of the descending aorta (DAo) or the ascending aorta (AAo). This hinders the standardization of the PT:Ao ratio in CT and, to the authors knowledge, there are no clinical studies comparing these two techniques. Therefore, the aim of this study was to determine the usefulness of the CT through the study of the PT:Ao ratio to detect presence and severity of PH in dogs with HWD, as well as to determine the concordance between PT:DAo and PT:AAo ratios.

## 2. Materials and Methods

### 2.1. Studied Animals

A cross-sectional and prospective study was developed using 59 dogs referred to the Cardiorespiratory Service of the Veterinary Teaching Hospital of the University of Las Palmas de Gran Canaria (Canary Islands, Spain), between September 2020 and July 2021. All dogs were tested for the presence of circulating *D. immitis* antigens by using a commercial immunochromatographic test kit (Urano test Dirofilaria^®^, Urano Vet SL, Barcelona, Spain). Dogs were further divided into groups: 30 (50.8%) healthy dogs and 29 (49.2%) heartworm-infected dogs. Inclusion criteria for healthy dogs were: a negative result to the antigen test, no record of previous disease, and absence of signs of cardiorespiratory disease based on anamnesis, physical examination, two-view thoracic radiographs and echocardiographic examination. Inclusion criteria for dogs with HWD were: the presence of *D. immitis* circulating antigens and never having received treatment for HWD. The study was performed before starting the adulticide treatment; however, nine heartworm-infected dogs were previously treated with pimobendane, benazepril, furosemide, torsemide, spironolactone, prednisolone, budesonide, sildenafil or doxycycline, after verifying echocardiographically that the PH persisted unchanged once the animals were stabilized [4]. A complete record was kept for each animal, including identification (age, sex, and breed), clinical history, and demographic data. The dog owners were informed and consented to participate in the study.

### 2.2. Echocardiography

Standardized transthoracic echocardiograms were performed in right and left lateral recumbent positions using ultrasound equipment (Vivid Iq, General Electric, Boston, USA) with spectral probes, color Doppler and multi-frequency (2.5–10 MHz). The dogs were conscious, without use of anesthesia and under electrocardiographic control throughout the test. For each measurement, six continuous cardiac cycles were recorded. All echocardiographic recordings were performed by a single cardiologist.

The diagnosis of the presence and severity of PH was echocardiographically determined, based on the determination of the right pulmonary artery distensibility index (RPAD index). Moderate to severe PH was considered when the RPAD index was <29.5%, as previously described and validated in dogs with HWD [5,15,16]. Complete echocardiograms were performed in all cases, and included the peak of tricuspid regurgitation velocity to obtain the tricuspid regurgitation pressure gradient (TRPG), the relationship between the diameter of the main pulmonary artery and the diameter of the aortic root (MPA:Ao), the right ventricular outflow tract velocity, and the relationship between acceleration times (AT) and ejection times (ET) using the AT:ET ratio. Furthermore, worm load (classified from 1—low burden, to 4—high burden) was assessed by the presence of visible worms in the pulmonary arteries and the right heart chambers using the guidelines previous described [17].

### 2.3. Computed Tomography

CT examinations were performed by a 16-slice multi-detector row scanner (Toshiba Astelion, Toshiba Medical System, Madrid, Spain). In all cases, a standard anesthesia protocol was used, based on propofol (up to 4 mg/kg) administered after alfaxalone IM (1 mg/kg) and butorphanol IM (0.2 mg/kg). Subsequently, isoflurane and 100% FiO_2_ were maintained with continuous monitoring of vital parameters (oxygen saturation, blood pressure, temperature, heart rate, etc.). Dogs were kept in sternal recumbency and postcontrast positive-pressure CT scans were performed after administration of 2.0 Ml/kg of iobitridol (300 mgI/Ml) through a catheter placed in the left or right cephalic vein. The study protocol included hyperventilation-induced apnea and positive pressure ventilation at 10 mm Hg. Helical cross-sectional images were acquired using a 0.625 slice thickness at 120 kVp, 200 Ma, 1 s tube rotation time, 512° ø 512 array size, and 0.9375: 1 pitch. To obtain optimal CT images and better assess mediastinal structures, a soft tissue window setting was used by adjusting the window widths (WW = 360) and window levels (WL = 60).

For the evaluation of the PT:Ao ratio, two different measurements were made to obtain the relationship between the PT and the ascending (PT:AAo) and descending (PT:DAo) portions of the aorta. The images of the soft tissue algorithm were used and the ratios were determined following previously established protocols [9,10]. To this aim, the cross-sectional image in which the diameter of the PT was widest was selected. The measurement of the PT was performed by tracing a transverse axis over the widest diameter of the PT, ventrally to the widening and branching in the left and right pulmonary arteries. Furthermore, the dimensions of the short axis of the DAo and the AAo were also measured. The diameters of DAo, AAo and PT were measured on the same image, so that all structures were in the same phase of the cardiac cycle. In all cases, the inner diameters of the vascular lumen was measured, not considering the vascular walls (Figure 1). The DICOM CT images were transferred to a computer and analyzed using an imaging software program (OsirixTM v. 5.6, Geneva, Switzerland). The interobserver variability of PT:Ao image acquisition and PT:DAO/PT:AAo measurement was evaluated separately in 10 dogs by two different cardiologists trained and supervised by a radiology consultant, obtaining comparable values. Measurements for all dogs were performed by a single cardiologist.

### 2.4. Statistical Analysis

Statistical analyses were performed using commercially available software (BM SPSS Statistics 25.0, New York, NY, USA). Continuous variables were reported as the median and standard deviation and categorical data as proportions. The Shapiro–Wilk test was applied to evaluate the normality of the distribution of continuous variables. Differences in continuous variables between groups were determined by ANOVA, Kruskal–Wallis and Chi^2^ test analyses. When significant differences were identified, post hoc pairwise comparisons were made using the Pearson P test with Bonferroni corrections (for non-normal data). The intraclass correlation coefficient was calculated to measure the concordance between PT:DAo and PT:AAo. The results of the statistical procedures were also graphed by scatter plot. A simple linear regression was performed between the RPAD index values and the other variables studied to identify the best one-variable model, and a regression analysis of all subsets was performed with a maximum improvement of R2 as a selection criterion. Receiver operator characteristic curve (ROC) analyses were performed to determine the optimal cut-off values for the prediction of the RPAD index being < 29.5% (moderate or severe hypertension). For all results, *p* < 0.01 was considered statistically significant.

## 3. Results

A total of 59 animals (31 males and 28 females) of 22 different breeds were analyzed, with a mean age of 8.72 years and a mean weight of 18.44 kg. Based on the RPAD index, all healthy animals (Group A, n = 30) were normotensive. Dogs with heartworm were further divided into two groups: normotensive dogs (Group B, n = 11) and dogs with PH (Group C, n = 18). There were no statistically significant differences in sex, age, and weight between the three groups studied (Table 1). Right-sided congestive heart failure was only present in dogs from Group C. Moreover, the presence of respiratory symptoms was significantly increased in dogs from Group C. Obviously no parasites were found in the control dogs (Group A). Burdens were similar in the dogs with heartworm (Groups B and C), being slightly higher in dogs from Group C.

Figure 2 illustrates the PT:DAo and PT:AAo ratios obtained for each group. The ANOVA tests indicated that there were significant differences between the values of the studied parameters and the groups of dogs. Bonferroni post hoc tests for multiple comparisons indicated that for PT:DAo, PT:AAo, MPA:Ao, AT:ET, and TRPG, there were significant differences between Group C when compared to Groups A and B. Furthermore, the results of Group A and Group B did not differ significantly. The median of the AT:ET ratio was significantly lower in dogs from Group C, while the median of the other parameters (TRPG, MPA:Ao, PT:DAo, PT:AAo) was higher in dogs from Group C. Moreover, statistically significant differences were found for the RPAD index between the three groups, being lower in Group C and higher in Group A (Table 1).

To measure the concordance between PT:DAo and PT:AAo, the Pearson correlation was performed between the two ratios, resulting in a very high correlation: 0.976 (*p*-value 0.000). Moreover, since correlation does not imply concordance, the intraclass correlation coefficient was also calculated. The results showed that both absolute agreement and consistency were excellent, with values > 0.95. Therefore, the concordance of both measures was demonstrated.

Linear regressions were performed through Pearson correlations. The R2 of the regressions that allowed comparison of the quality of the studied measures with the RPAD index were obtained. The method that best correlated was AT:ET (R2 > 0.8), followed by PT:DAo and TRPG with R2 > 0.7, and PT:AAo and MPA:Ao with R2 > 0.65 (Table 2). Finally, five ROC curves were determined for each method in which the event to be studied was having PH (RPAD index < 29.5%). The AUCs of all five parameters were excellent (>0.95) (Table 3). For PT:DAo, any value ≥ 1.111 concurred with PH in 95.1% of cases, and a value < 1.111 concurred with normotension in 100% of cases. For PT:AAo, any value ≥ 1.057 corresponded to PH in 100% of cases and a value < 1.057 corresponded to normotension in 100% of cases. Cut-off values, sensitivity, and specificity for the other studied parameters are shown in Table 3.

## 4. Discussion

PH is a frequently occurring and severe condition in dogs with HWD. It is a consequence of the presence of adult parasites of *D. immitis*, which produce severe lesions in the pulmonary arteries that lead to proliferative endarteritis. These endothelial alterations start as soon as the worms arrive at the pulmonary arteries and the chronic consequences cause a marked decrease in quality of life and life span [1,18,19]. This fact is observable in the results of this study, in which dogs with PH showed significantly higher clinical signs of disease and presence of congestive heart failure. Moreover, it has been reported that endarteritis may not be reversible once the parasites have been eliminated and chronic PH has been described in dogs up to 10 months after completion of adulticide treatment [4,20]. Therefore, it is extremely important to find different diagnostic methods for its detection. The damage produced by *D. immitis* to the endothelium cause a reduction in elasticity of the pulmonary arteries, being associated with an impaired distensibility of the vessel [20]. For this reason, the RPAD index has been described as a valuable echocardiographic method to detect and assess severity of PH in dogs with HWD, compared to other echocardiographic variables including AT:ET, TRPG y MPA:Ao [3,4,5,20]. The results of the present study confirmed the utility of the RPAD index as a gold standard for the echocardiographic detection of PH in dogs with HWD. Furthermore, the RPAD index was the only studied parameter that is significantly modified for dogs with heartworm but the absence of HP or the presence of mild PH (RPAD index ≥ 29.5%), compared to healthy dogs, which demonstrates its utility as an early marker of endothelial damage in this pathology.

Moreover, the results of the additional echocardiographic techniques used to compare and detect PH have demonstrated their usefulness in determining the presence of PH, finding satisfactory strong correlations compared to the RPAD index [5,16,17]. The proportions observed and the values obtained in the present study show similar data to results published in other echocardiographic investigations, producing promising results for the non-invasive diagnosis of PH in HWD [21,22].

The evaluation of PH in dogs can be a challenge in veterinary medicine and the use of new useful and easily repeatable diagnostic methods is of utmost importance to confirm the finding of PH. Studies carried out during the last decade have reported the growing use of thoracic CT for the diagnosis and management of dogs with cardiorespiratory diseases that may predispose to PH, including heartworm. Concerning the latter, previous studies have shown the usefulness of the CT study to diagnose pulmonary arterial and parenchymal changes as a consequence of heartworm infection [23,24]. Thus, the diagnostic utility of CT-derived measurement of the PT and Ao diameter to predict the presence of PH in HWD may be advantageous, as shown in the results of this study.

Furthermore, the intraclass correlation study developed to determine the usefulness of measuring the ratio using the two portions of the thoracic aorta, ascending and descending, reported that both methods were equally valid to study the presence of PH by CT. The results showed lower mean values when using the ascending portion (PT:AAo ratio) as a result of larger aortic measurement diameters in this portion compared to the descending portion (PT:DAo). Based on the clinical experience of the authors, the values obtained through the PT:AAo ratio are easily comparable to the echocardiographic MPA:Ao measure, since both measures are determined at the same vascular level. Moreover, both techniques are simple and reproducible. The presence of adult parasites was not observed in any of the animals studied, demonstrating that this technique is not useful for quantifying or diagnosing the presence of adult parasites in the pulmonary arteries of infected dogs.

The study of the PT:Ao ratio via CT in humans has previously been reported [12,13,14], in which the increase in the diameter of the pulmonary trunk compared with the descending and the ascending portions of the thoracic aorta have been shown to have a strongly predictive value of PH, and cut-off points were established to determine the presence of even mild PH [14]. In veterinary medicine, recent studies have also been published on the PT:Ao ratio with the objective of establishing physiological measures in healthy animals and to establish cut-off points to diagnose the presence and severity of PH [9,10,11]. In this study, unlike the RPAD index, the PT:Ao ratio did not show significant differences between healthy dogs and dogs with heartworm without hypertension or mild PH. These results are similar to those reported in previous investigations, which showed a high degree of overlap between PT:Ao ratios of the non-PH and mild-PH groups [9,10,11].

On the other hand, the results displayed a great usefulness of this ratio to detect moderate to severe HP in dogs with heartworm, showing cut-off values with high sensitivity and specificity for PT:DAo and PT:AAo. The cut-off point to diagnose moderate or severe PH reported for PT:DAo in this study (≥1.111) was slightly lower than previously reported by other authors (> 1.4) in dogs with PH of undescribed etiology [10]. Moreover, the mean PT:AAo and PT:DAo obtained for healthy dogs were slightly lower than the values presented in previous studies (1.11 ± 0.15 and 0.97 ± 0.12 for PT:AAo, and 1.26 ± 0.11 for PT:DAo) [9,10,11]. These differences could be due to the different methodologies established to determine the presence or absence of PH in the studies mentioned. In addition, differences in cross-sectional measurements of vascular diameters from previously published articles may lead to small differences in the determination of the PT:Ao ratio. Moreover, an optimal PT:Ao ratio thresholds for the diagnosis of PH have been poorly defined and the diagnostic value for evaluating PH using both the ascending and descending parts of the Ao have not been accomplished in HWD, so the results of this study are of great novelty and clinical usefulness.

As has been previously reported in dogs with pulmonary arterial hypertension produced for other precapillary causes, an increase in the relationship between PT and Ao was associated with the severity of PH diagnosed by echocardiography [9,10,11]. These findings are consistent with the results of the present study, which showed an excellent correlation between the RPAD index and the PT:Ao ratios in dogs with heartworm. However, the lack of utility to detect mild PH is an important limiting factor, since the detection of early endothelial damage in heartworm is important for gaining good control and implementing early therapeutic measures for a better prognosis.

The main limitation of the present study was the size of the canine population analyzed, and the cross-sectional anatomy between dogs’ size, weight and breed, so further studies with higher number of animals are warranted. In addition, the pulmonary arterial pressure was not directly measured by right heart catheterization, but instead used the RPAD index as the gold standard in the measurement of HP in dogs with *D. immitis*.

The phase of respiration must be considered if measurement of PT:Ao through CT is anticipated because a statistically significant difference was seen between the vascular measurements on the expiratory CT scans when compared to both the pre- and postcontrast inspiratory CT protocols [9]. For this study, based on recent publications, the use of postcontrast protocols was chosen [10,11]; however, the respiratory cycle could not be defined and the influence of inspiration and expiration on the size of the PT and the portions of Ao was not analyzed. Uniform tomographic studies were not obtained in all cases, so multiplanar reconstructions were not used to determine vascular measurements. The fact that dogs were anesthetized during CT is an important limitation, but was mandatory and reflects real clinical practice. Anesthesia alters hemodynamics with potential repercussions on the measures of CT indices of PH tested in this study. Premedication based on alfaxalone and butorphanol has proven to be hemodynamically very stable in other cardiorespiratory disorders, and the use of propofol and isoflurane makes it possible to standardize the method used, since they are drugs widely used in anesthetic induction and maintenance in veterinary practice [12]; therefore, the results obtained are subject to the particular use of the anesthetic protocol used.

## 5. Conclusions

In conclusion, the results of the present study highlight the potential utility of the PT:Ao ratio, using PT:AAo or PT:DAo PT:Ao ratios, to diagnose moderate to severe PH in dogs with heartworm. Although the RPAD index remains the gold standard for the diagnosis of PH in these dogs, dogs undergoing CT for any other clinical indication could benefit from the study and standardization of the PT:Ao ratio. However, further studies with a larger number of animals are needed to evaluate and identify whether the TC parameters studied have clinical utility for the early diagnosis of PH in dogs with heartworm.

## Figures and Tables

**Figure 1 animals-12-02441-f001:**
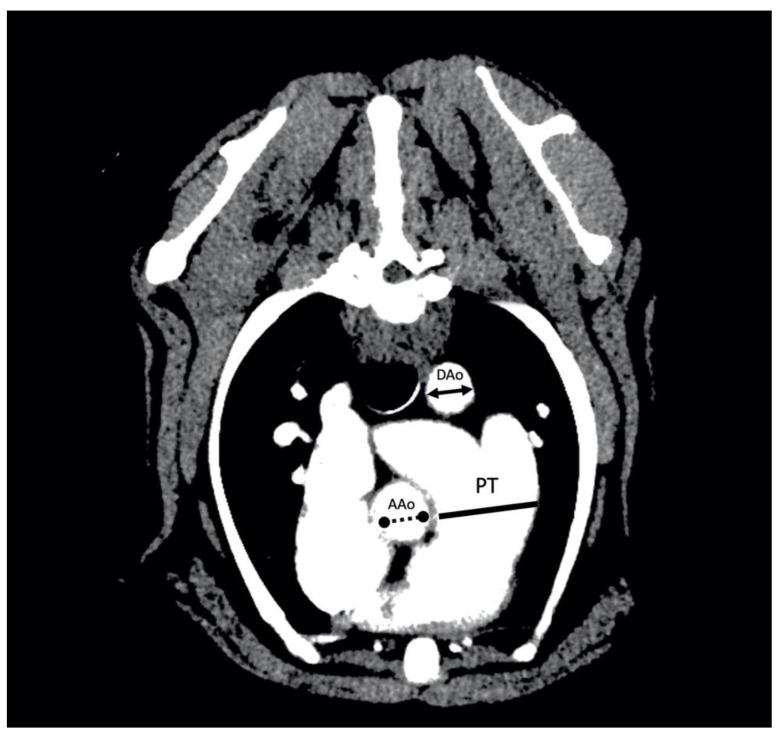
Arterial-phase postcontrast transverse thoracic angiogram CT image at the level of the pulmonary trunk (PT) in a 16 kg male beagle with an RPAD index of 17%. The PT to aorta ratio measurement technique consists of the measure of the maximum diameter of the PT measured immediately ventral to the bifurcation into left and right pulmonary arteries (solid line) and the measure of the short axis of the diameter of the descending part of the aorta (DAo) (solid line double arrow) or the short axis of the diameter of the ascending part of the aorta (AAo) (no solid line double arrow). The PT:DAo ratio of this dog was 1.72 and the PT:AAo ratio was 1.47.

**Figure 2 animals-12-02441-f002:**
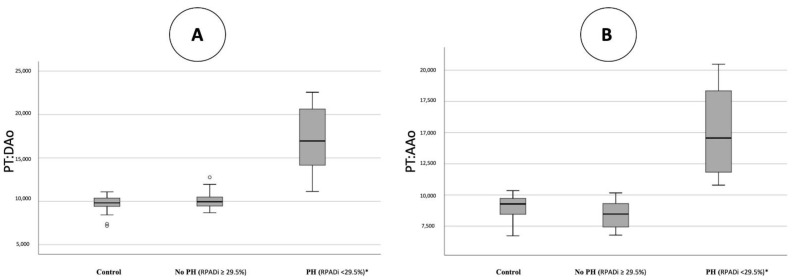
Scatterplots illustrating pulmonary trunk to descending aorta ratio (PT:DAo) (**A**) and pulmonary trunk to ascending aorta ratio (PT:AAo) (**B**), obtained in healthy dogs (control), in dogs with heartworm and no presence/mild pulmonary hypertension (RPADi ≥ 29.5%), and dogs with heartworm and moderate to severe pulmonary hypertension (RPADi < 29.5%). The box plots represent median (solid horizontal lines within boxes), 25th, and 75th percentiles (boxes) and minimum and maximum values (whiskers). Statistically significant differences (*) were observed between control and RPADi ≥ 29.5% dogs when compared to RPADi < 29.5% dogs.

**Table 1 animals-12-02441-t001:** Clinical characteristics and parameters of the studied dogs (n = 59).

Clinical Parameters	All Dogs (n = 59)	Control (n = 30)	RPADi ≥ 29.5% (n = 11)	RPADi < 29.5% (n = 18)	*p*-Value
**Body weight (kg)**	18.44 ± 11.84	16.14 ± 11.15	24.35 ± 12.30	18.65 ± 12.03	0.232 ^b^
**Age (years)**	8.72 ± 3.46	8.86 ± 3.94	8.33 ± 1.85	8.73 ± 3.51	0.915 ^a^
**Female: number (%)**	28 (47.5%)	14 (46.67%)	4 (36.36%)	10 (55.56%)	0.59 ^c^
**Respiratory symptom (%)**	14 (23.73%)	0 (0%)	3 (27.27%)	11 (61.11%)	0.00 ^c^
**Right-sided CHF (%)**	7 (24.13%)	0 (0%)	0 (0%)	7 (38.9%)	0.00 ^c^
**TRPG (mmHg)**	27.02 ± 34.51	7.88 ± 6.75	10.24 ± 7.93	69.15 ± 35.22	0.00 ^a^
**RPAD index (%)**	34.4 ± 10.1	41.9 ± 4.9	34.9 ± 2.0	21.8 ± 6.3	0.00 ^a^
**MPA:Ao**	1.08 ± 0.23	0.95 ± 0.07	0.947 ± 0.07	1.357 ± 0.21	0.00 ^a^
**PT:DAo**	1.21 ± 0.41	0.97 ± 0.09	1.02 ± 0.12	1.72 ± 0.39	0.00 ^a^
**PT:AAo**	1.08 ± 0.363	0.90 ± 0.09	0.84 ± 0.11	1.53 ± 0.34	0.00 ^a^
**AT:ET**	0.34 ± 0.09	0.39 ± 0.05	0.37 ± 0.06	0.22 ± 0.06	0.00 ^a^
**Parasite burden (1–4)**	2.27 ± 1.34	0	2.09 ± 1.47	2.56 ± 1.22	0.00 ^c^

Legend: Results for body weight, age, tricuspid pressure gradient (TRPG), right pulmonary artery distensibility index (RPAD index), main pulmonary artery to aorta ratio (MPA:Ao), pulmonary trunk to descending aorta ratio (PT:DAo), pulmonary trunk to ascending aorta ratio (PT:AAo), right acceleration time to ejection time ratio (AT:ET) and parasite burden are expressed as mean ± standard deviation. Results for female, respiratory symptoms, and right sided congestive heart failure (CHF) are expressed as n (%). *p* values: ^a^
*p* < 0.01 = ANOVA; ^b^
*p* < 0.01 = Kruskal–Wallis; ^c^
*p* < 0.01 = Chi-squared test.

**Table 2 animals-12-02441-t002:** Results of simple regression analyses for the prediction of RPAD index.

Method	Pearson Corralation	R^2^	*p*-Value	CI 95%	Regression Equation
**PT:DAo**	0.845 **	0.709	0.000	(−0.244;−0.174)	RPADIndex = 0.597 − 0.209 * PT/DAo
**PT:AAo**	0.824 **	0.674	0.000	(−0.272;−0.188)	RPADIndex = 0.593 − 0.230 * PT/AAo
**AT:ET**	0.898 **	0.803	0.000	(0.812;1.054)	RPADIndex = 0.030 + 0.933 * AT:ET
**MPA:Ao**	0.830 **	0.684	0.000	(−0.433;−0.301)	RPADIndex = 0.738 − 0.367 * MPA/AO
**TRPG**	0.847 **	0.712	0.000	(−0.003;−0.002)	RPADIndex = 0.412 − 0.002 * TRPG

Legend: PT:DAo: pulmonary trunk to descending aorta ratio; PT:AAo: pulmonary trunk to ascending aorta ratio; AT:ET: right acceleration time to ejection time ratio; MPA:Ao: main pulmonary artery to aorta ratio; TRPG: tricuspid pressure gradient; CI 95%: 95% confidence interval; R2: coefficient of determination; RPADIndex: right pulmonary artery distensibility index. On the one hand, “**” means that the correlation is significant at the 0.01 level (bilateral). On the other hand, “*” is used in regression equations as a multiplication sign.

**Table 3 animals-12-02441-t003:** Sensitivity (Se), specificity (Sp), and Youden index of cut-off points of the studied parameters/measures to predict PH estimate based on the right pulmonary artery distensibility index (RPAD index) < 29.5%.

Method	AUC	CI 95%	Cut-Off	Se	Sp	Youden Index (Se + Sp-1)
**PT:DAo**	0.993	(0.980; 1.000)	≥1.111	0.951	1.000	0.951
**PT:AAo**	1.000	(1.000; 1.000)	≥1.057	1.000	1.000	1.000
**AT:ET**	0.983	(0.958; 1.000)	≤0.325	0.854	1.000	0.854
**MPA:AO**	1.000	(1.000; 1.000)	≥1.098	1.000	1.000	1.000
**TRPG**	0.999	(0.994; 1.000)	≥25.874	0.976	1.000	0.976

Legend: PT:DAo: pulmonary trunk to descending aorta ratio; PT:AAo: pulmonary trunk to ascending aorta ratio; AT:ET: right acceleration time to ejection time ratio; MPA:Ao: main pulmonary artery to aorta ratio; TRPG: tricuspid pressure gradient; AUC: area under receiver operating characteristic curve; CI 95%: 95% confidence interval; Se: sensitivity; Sp: specificity.

## Data Availability

The raw data supporting the conclusions of this article will be made available by the authors, without undue reservation.

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
