# Peer review of "Evaluation of Pulmonary Hypertension in Dogs with Heartworm Disease Using the Computed Tomographic Pulmonary Trunk to Aorta Diameter Ratio"

_animals, 2022, doi:10.3390/ani12182441_

Round 1
Reviewer 1 Report
The reviewer found the topic and the idea interesting for the clinical practice.
However, there are some points that should be clarified before considering the manuscript publication.
1.considering the sample of the animals included, in line 88 there is the mention to the inclusion of 9 animals previously treated. Authors should justify in detail such choice and clarify if the previous treatment could have influenced the various measurements and, in such case, consider to exclude the dogs from the study and repeat the examination on a more homogeneous sample in order to avoid such biases.
2. in line 122 there is the mention to the hyperventilation-induced apnea. Since the respiratory cycle could not be defined and the influence of inspiration and expiration on the size of the PT and the portions of Ao was not analyzed, why the authors did not perform the apnea through a mechanical ventilator? Do the author believe that this could have helped to better standardize the protocol and the results?
3. in line 329, authors states, correctly, that anesthesia alters hemodynamics with potential repercussions on measures of CT indices of PH tested in this study; authors should better clarify the choice of the drugs used for this protocol potential influence of the anesthetics protocol herein described on the results obtained.
Also, they should discuss which drugs could potentially influence the patient hemodynamics.
Furthermore, since for CT anesthesia is necessary, authors should better clarify in which clinical cases patients could benefit more from CT more than US to detect PH, since it has been observed that CT ratios are useful to diagnose moderate to severe PH in dogs and thus potentially limiting the anesthesia safety in such patients.
4. In line 258, authors assess that “easily repeatable diagnostic methods is of outmost importance to confirm the finding of PH”; the reviewer totally agrees on this point, and suggests to repeat the measurements with at least two blinded operators in order to evaluate the reproducibility of such measurements.
5. During the CT measurements, was the cardiologist supervised or trained by a radiologist?
Reviewer 2 Report
A very well-written and interesting study, congratulations!! Some suggestions/ modifications:
- In order to obtain better mediastinal structures, a slice thickness greater than 1 mm must be used. These 0,625 mm slices are preferred for lung imaging, not for soft tissue. A soft tissue reconstruction can be obtained as a second reconstructed series with the original data in thinner sections. Line 132: The axis obtained of the PT is transverse and not longitudinal (respect to the animal plane), as it is in the Ao (DAo and AAo). In figure 1 the trace of the PT is not correctly perpendicular, and it may alter the value (maybe not statistically significant). It must be clarified that is the inner diameter that is measured (without the vessel walls) in any case (Ao and PT).
- Lines 172-174: Can be considered “healthy” dogs those with Dirofilaria infestation, even when it is very low? Healthy animals were supposed to be negative at this test --> this sentence must be modified to clarify the paragraph.
- Paragraph line 286: Differences in ratio values of the present study vs previous studies must be in the method for obtaining TP cross sectional diameter, as these differences are not large between them.
- Question? Were adult worms visible in postcontrast CT images in any case? This could be a good conclusion also.
- Anesthesia vs sedation variations in these parameters can also be investigated.
Reviewer 3 Report
Dear authors,
You addressed an important topic in your manuscript, canine dirofilariasis being one of the main parasitic diseases that can lead to the death of our beloved pet.
The manuscript is well-written, but a major revision, in my view, is needed.
I carefully read your published article:
Serrano-Parreño, B.; Carretón, E.; Caro-Vadillo, A.; Falcón-Cordón, Y.; Falcón-Cordón, S.; Montoya-Alonso, J.A. Evaluation of pulmonary hypertension and clinical status in dogs with heartworm by Right Pulmonary Artery Distensibility Index and other echocardiographic parameters. Parasit Vectors 2017, 10, 106-112.
By the way, you referenced it twice, as [3] and [6].
Apart from this minor error, this new manuscript is worked and written by a similar authors team (three common authors for both papers) from the same institution. I understand that a strong team of veterinarians was developed at the Internal Medicine, Veterinary Medicine, and Therapeutic Research Group from the Spanish faculty. I also realize you initially (2017) echocardiographically evaluated RPAD Index as "an objective and supportive test in the monitoring and evaluation of PH in the heartworm-infected dog, and show a potential diagnostic value for the detection of PH in asymptomatic animals".
In the submitted manuscript, you evaluate a new technique, computed tomography, in the same field, evaluation of pulmonary hypertension in dogs with heartworm disease. Echocardiography was the evaluation method in the first study, whereas now, apart from this technique, you also tested computed tomographic. The studied animals differ within the two papers, the published one and the currently submitted. Still, in this submitted manuscript, you used echocardiography again, determining the same parameters: RPAD index, TRPG or TR, AT:ET ratio or AT/ET ratio, and so on.
My problem is the following: under the circumstances of the use and detailed description of echocardiography again, and the results obtained, even if other dogs are used, and even if you have referenced your published article, all echocardiography-related data in this manuscript can be considered self-citation.
I think the best option to avoid this issue is to remove all echocardiography-related data in the submitted manuscript from the Materials and Methods and Results sections, keeping only references from the Discussion section.
I understand the study is more complete if includes results of echocardiography, but this is my position regarding self-citations.
Round 2
Reviewer 1 Report
Dear authors,
thank you for your detailed reply to the comments provided.
In my opinion, the paper is now acceptable in the present form.
Reviewer 3 Report
I fully understood your answer regarding the repetition of some elements previously presented in already published articles. Indeed, this manuscript would have been difficult to understand without their re-presentation. I consider it publishable in its current form.
Congratulation!